# Integral Analysis of Liquid-Hot-Water Pretreatment of Wheat Straw: Evaluation of the Production of Sugars, Degradation Products, and Lignin

Sebastian Serna-Loaiza [1,*], Manuel Dias [2,3], Laura Daza-Serna [1], Carla C. C. R. de Carvalho [2,3] and Anton Friedl [1]

1    Institute of Chemical, Environmental and Bioscience Engineering, Technische Universität Wien, 1060 Vienna, Austria; laura.serna@tuwien.ac.at (L.D.-S.); anton.friedl@tuwien.ac.at (A.F.)
2    iBB—Institute for Bioengineering and Biosciences, Department of Bioengineering, Instituto Superior Técnico, Universidade de Lisboa, Av. Rovisco Pais, 1049-001 Lisboa, Portugal; manuel.francisco.nunes.dias@tecnico.ulisboa.pt (M.D.); ccarvalho@tecnico.ulisboa.pt (C.C.C.R.d.C.)
3    Associate Laboratory i4HB—Institute for Health and Bioeconomy, Instituto Superior Técnico, Universidade de Lisboa, Av. Rovisco Pais, 1049-001 Lisboa, Portugal
*    Correspondence: sebastian.serna@tuwien.ac.at; Tel.: +43-1-58801-166262

**Abstract:** Developing sustainable biorefineries is an urgent matter to support the transition to a sustainable society. Lignocellulosic biomass (LCB) is a crucial renewable feedstock for this purpose, and its complete valorization is essential for the sustainability of biorefineries. However, it is improbable that a single pretreatment will extract both sugars and lignin from LCB. Therefore, a combination of pretreatments must be applied. Liquid-hot-water (LHW) is highlighted as a pretreatment for hemicellulose hydrolysis, conventionally analyzed only in terms of sugars and degradation products. However, lignin is also hydrolyzed in the process. The objective of this work was to evaluate LHW at different conditions for sugars, degradation products, and lignin. We performed LHW at 160, 180, and 200 °C for 30, 60, and 90 min using wheat straw and characterized the extract for sugars, degradation products (furfural, hydroxymethylfurfural, and acetic acid), and lignin. Three conditions allowed reaching similar total sugar concentrations (~12 g/L): 160 °C for 90 min, 180 °C for 30 min, and 180 °C for 60 min. Among these, LHW performed at 160 °C for 90 min allowed the lowest concentration of degradation products (0.2, 0.01, and 1.4 g/L for furfural, hydroxymethylfurfural, and acetic acid, respectively) and lignin hydrolysis (2.2 g/L). These values indicate the potential use of the obtained sugars as a fermentation substrate while leaving the lignin in the solid phase for a following stage focused on its extraction and valorization.

**Keywords:** biorefineries; hemicellulose; lignin; liquid hot water; wheat straw

## 1. Introduction

Developing sustainable biorefineries is urgent to address the fossil-based environmental impacts and transition to a sustainable model [1]. Lignocellulosic biomass is a critical renewable feedstock to achieve this purpose. However, the resistance to degradation of biomass still represents a challenge to effectively fractionate its different components: cellulose, hemicellulose, and lignin [2,3]. Besides, the overall valorization of biomass is essential for biorefinery sustainability, including the valorization of hemicellulose and lignin [4]. This would provide two different platforms for value-added products, and cellulose could still be valorized either as a fiber or through enzymatic conversion [5].

In many cases, the ultimate goal of the pretreatment stage is to deconstruct the lignin−hemicellulose complex and increase cellulose availability. However, the three fractions have valorization potential, and pretreatments should be designed to use hemicellulose and lignin. One of the major challenges in this regard is the simultaneous valorization of sugars from the hemicellulose fraction and lignin valorization [6].

Multiple pretreatment clusters focused on the deconstruction of the lignocellulosic matrix have been proposed. Each of these technologies has advantages and disadvantages [7]. Liquid Hot Water (LHW) uses only water as a reagent, but on the other hand, it requires high amounts of energy for heating and cooling. Similarly, steam explosion uses only water, but it requires higher pressures than LHW, which implies higher energy consumption. Organosolv pretreatment uses an aqueous solution of an organic solvent, which allows solubilizing both hemicellulose and lignin. However, the process feasibility is strictly related to the recovery of the solvent in a distillation setup. Alkaline pretreatments are also used for a similar purpose, such as the Organosolv process. However, the concentrations of alkalis, such as sodium hydroxide and sodium carbonate, lead to higher corrosion, and alkali recovery is a challenging task. Among the pretreatments, LHW stands out for hemicellulose hydrolysis, using water as a reactant without further input of acids/bases. Besides, this pretreatment is auto-catalyzed by the acetic acid formed from the acetyl groups released from the hemicellulose backbone [8,9].

However, a single pretreatment to obtain simultaneously high sugar and lignin yields has not been found yet, and a combination of pretreatments is usually applied [5]. Previous studies have evaluated different combinations of pretreatments. Xia et al. (2020) evaluated LHW followed by sodium carbonate-oxygen pretreatment to improve the reed enzymatic saccharification [10]. Neves et al. (2016) and Rocha et al. (2012) studied steam explosion followed by alkaline pretreatment of sugarcane bagasse [11,12]. Tian et al. (2019) combined LHW with mechanical extrusion from rigid hardwood [13]. Wang et al. (2012) combined fungal treatment with LHW of white poplar [14]. Serna-Loaiza et al. (2021) combined LHW and Organosolv to produce sugars from the hemicellulose and hydrolyze the lignin from wheat straw. This work found that performing LHW before Organosolv increased the lignin extraction yield by 1.6-times (from 7 to 11 g/L in the OS-stage) [15]. Other authors have evaluated the same configuration, LHW followed by Organosolv, for different raw materials such as hazelnut shells [16] and corncobs [17].

This research trend shows the potential of pretreatment combinations as a strategy to valorize the different fractions of lignocellulosic biomass integrally. Nonetheless, a combination of pretreatments implicates analyzing every stage in terms of the different components to be extracted, as this will provide more detailed information to establish the mass balance of the process. Traditionally, LHW has been analyzed solely as a stage for hemicellulose hydrolysis, as this is its primary purpose. Hence, it is analyzed only in terms of sugar and respective degradation products. However, LHW also hydrolyses part of the lignin present in the feedstock. Therefore, it is necessary to analyze LHW from an integral perspective, including lignin hydrolysis. This information is essential to design subsequent pretreatment stages, mainly focused on the lignin fraction. Therefore, in the present work, LHW of wheat straw at multiple temperatures (160, 180, 200 °C) for different times (30, 60, 90 min) was performed and the extracts were analyzed for sugars, degradation products, and lignin concentration. The main goal of this study was to gain a better insight into which of the tested conditions allow obtaining a sugar hydrolysate with the highest sugar concentration and the lowest concentration of degradation products and extracted lignin. The lignin remaining in the solid phase could be extracted in a further process step.

## 2. Materials and Methods

### 2.1. Raw Material and Reagents

Wheat straw used in this work was harvested in 2019 (Margarethen am Moos, State of Lower Austria) and stored under dry conditions at room temperature. The straw was milled in a cutting mill, equipped with a 2 mm mesh. The raw material composition was 2.13%, 0.67%, 35.31%, 21.94%, 0.72%, 17.35%, 20.45%, and 1.09% (wt; dry basis) for arabinan, galactan, glucan, xylan, mannan, lignin, extractives, and ash, respectively [15]. The wheat straw used in this work corresponds to the same sample and batch as the one characterized in the cited study. In addition, it was selected as a test raw material due to its high content of hemicellulose and lignin, which is the main targeted lignocellulosic components for the

experimental section. In addition, this raw material has been significantly studied for both processes, LHW and Organosolv [15,18–22], which shows the relevance of understanding more deeply the influence of LHW, including the lignin determination. The moisture content was 7.16 wt %. Ultra-pure water (18 MΩ/cm) was used for the LHW. Standards for carbohydrates (arabinose, galactose, glucose, xylose, and mannose), acetic acid (99.7%), 2-furaldehyde (furfural, 99%), hydroxymethylfurfural (HMF, 99%), and sulfuric acid (98%) were purchased from Merck (Darmstadt, Germany).

## 2.2. Process Condition and Description

LHW is generally performed in the temperature range between 160–240 °C during 30–120 min [23]. The present study chose low to intermediate values within these ranges: temperatures tested were 160, 180, and 200 °C, and selected times were 30, 60, and 90 min. Each combination of conditions was performed in triplicate. The time corresponds to the holding time, meaning that once the reactor reaches the desired temperature, the temperature is maintained constant for the fixed time.

LHW was carried out in a stainless steel pressurized autoclave (Zirbus, HAD 9/16, Bad Grund, Germany) with a capacity of 1 L and maximum temperature and pressure of 250 °C and 60 bar, respectively. The autoclave was equipped with a controller registering every second the inlet temperature of the reactor, which was used to calculate the severity factor. The reactor stirring speed was set to 200 rpm. The reactor was heated to the operating temperature and rapidly cooled down to room temperature after the defined holding time. At the end of the process, the sample was collected, and the solid and liquid fractions were separated using a hydraulic press (Hapa, HPH 2.5) at 200 bar. The liquid fraction was centrifuged (Sorvall, RC 6+) at 24,104× $g$ for 20 min, and the supernatant was stored at 5 °C until further analysis. The density of the extract was determined using a density meter (DE45 DeltaRange, Mettler Toledo, Columbus, OH, USA).

The initial wet mass of wheat straw was approximately 32.31 g (corresponding to 30 g dry weight). The solid/liquid ratio was 1 g of dry solid per 11 g of solvent (solid loading of 8.3 wt %), while the added water was 327.69 g (after subtracting the water in the wet straw). The severity factor ($R_0$) was calculated using Equation (1) considering the heating (from 100 °C to the set temperature), temperature holding according to the set time, and cooling (from the set temperature to 100 °C), as shown in Equation (2). $t$ is the time (min), $T(t)$ is the temperature (°C), and Total $R_0$ is the sum of the contribution of each stage to the severity factor. The constant "14.75" corresponds to an empirical parameter calculated assuming an overall reaction following first-order kinetics and Arrhenius relation of temperature [24]. This integral was solved by the trapezoidal rule shown in Equations (3) and (4) using the data collected by the reactor controller with at $\Delta t$ of 1 s.

$$R_{0,i} = \int_0^t \exp\left(\frac{T(t) - 100}{14.75}\right) dt \tag{1}$$

$$\text{Total } R_0 = \log_{10} R_{0,Heating} \tag{2}$$

$$\int_a^b f(t)dt \approx (b - a) * \frac{f(a) + f(b)}{2} \tag{3}$$

$$\int_0^{t_{op}} f(t)dt \approx \sum_{i=0}^{t_{op}} \left( \Delta t * \frac{\exp\left(\frac{T_t - 100}{14.75}\right) + \exp\left(\frac{T_{t+\Delta t} - 100}{14.75}\right)}{2} \right) \tag{4}$$

## 2.3. Product Characterization

The liquid fraction was characterized for sugars (monomeric and total), degradation products (furfural, HMF, and acetic acid), and lignin (acid-soluble and acid-insoluble). Sugars and degradation products were characterized according to the NREL/TP-510-42623 [25].

Monomeric sugars were analyzed using high-performance ion-exchange chromatography (HPIEC-PAD) (DionexTM ICS-5000, Thermo Scientific, Waltham, MA, USA) with deionized water as eluent. The extract was hydrolyzed with diluted sulfuric acid at 120 °C and analyzed the sugars as monomers; this corresponded to the total sugars. Oligomeric sugars were calculated as the difference between total and monomeric sugars. A sugar recovery standard was used to account for losses. Furfural, HMF, and acetic acid were determined using HPLC (LC-20A HPLC system, Shimadzu, Kyoto, Japan), with UV and RI detection, with a Shodex SH1011 analytic column at 40 °C with 0.005 M $H_2SO_4$ as mobile phase. The extract was dried and analyzed for lignin determination according to the NREL/TP-510-42618. Acid insoluble lignin (AIL) was determined gravimetrically, and acid-soluble lignin (ASL) by UV/VIS absorption at 205 nm using a Shimadzu UV-1800 spectrophotometer [26].

## 3. Results

First, we present the calculated severity factors for the different temperature and time combinations. Then, we present the results for sugars, degradation products, and lignin in this respective order. Finally, we analyze the three indicators to identify the conditions that provide the highest sugar concentration, with the lowest concentration of degradation products and lignin.

### 3.1. Calculated Severity Factor

Table 1 shows the calculated severity factors for the different combinations of temperature and time. Due to the definition of Equation (1), an increase in the holding temperature influences more the severity factor than an increase in time. For example, increasing the holding time from 60 to 90 min, at 160 °C, increased the severity factor by 0.16, whereas increasing the temperature from 160 to 180 °C (at 60 min holding time) increased the severity factor by 0.60. Besides, as the holding time at a given temperature increased, the heating and cooling contribution decreased. Additionally, the cooling contribution could be neglected since, for all performed experiments, the cooling time was $9.0 \pm 0.7$ min and contributed $2.4 \pm 0.7\%$ to the total severity factor. Between the replicates, the actual time and respective standard deviations show only one condition with an error percentage of 7.9% (200 °C and 60 min), while the rest had errors below 3.9%.

**Table 1.** Calculated severity factors for the performed LHW extractions.

| Temp (°C) | Aimed Holding Time (min) | Real Holding Time (min) | Severity Factor ($R_0$, min) | Contribution of Stages to the Severity Factor (%) | | |
|---|---|---|---|---|---|---|
| | | | | Heating | Holding | Cooling |
| 160 | 30 | $30.7 \pm 0.2$ | $3.38 \pm 0.02$ | 16.4 | 79.4 | 4.2 |
| | 60 | $61.0 \pm 0.3$ | $3.61 \pm 0.00$ | 6.5 | 91.0 | 2.4 |
| | 90 | $90.2 \pm 0.2$ | $3.77 \pm 0.01$ | 5.7 | 92.3 | 2.0 |
| 180 | 30 | $31.6 \pm 0.6$ | $4.05 \pm 0.02$ | 20.6 | 76.1 | 3.2 |
| | 60 | $60.7 \pm 0.3$ | $4.22 \pm 0.01$ | 7.6 | 90.2 | 2.2 |
| | 90 | $91.1 \pm 0.6$ | $4.38 \pm 0.01$ | 5.3 | 93.1 | 1.7 |
| 200 | 30 | $30.6 \pm 0.4$ | $4.60 \pm 0.04$ | 20.8 | 76.1 | 3.1 |
| | 60 | $60.5 \pm 0.3$ | $4.81 \pm 0.02$ | 11.9 | 86.3 | 1.8 |
| | 90 | $87.1 \pm 4.2$ | $4.97 \pm 0.02$ | 12.6 | 86.2 | 1.2 |

### 3.2. Sugar Content

Table 2 shows the sugar concentrations, both monomeric and total, for the performed LHW extractions. As can be observed, the sugars with higher concentrations in increasing order were arabinose, glucose, and xylose. Galactose and mannose showed lower concentrations in all extraction conditions. This trend is in accordance with the composition previously shown in Section 2 for the used wheat straw [15]: the glucoarabinoxylan poly-

mer is significantly more abundant than the galactomannan. We summed the respective C5 (arabinose and xylose) and C6 (galactose, glucose, and mannose) sugars to better represent the overall production of sugars at the different conditions.

**Table 2.** Sugar concentrations (monomeric and total) for the performed LHW extractions.

| T (°C) | Time (min) | Severity Factor | Concentration (mg/L) | | | | | | | | | |
| --- | --- | --- | --- | --- | --- | --- | --- | --- | --- | --- | --- | --- |
| | | | Arabinose | | Galactose | | Glucose | | Xylose | | Mannose | |
| | | | **M** | **T** | **M** | **T** | **M** | **T** | **M** | **T** | **M** | **T** |
| 160 | 30 | 3.38 | 422 ± 23 | 987 ± 53 | 18 ± 1 | 385 ± 15 | 63 ± 7 | 1014 ± 17 | 58 ± 6 | 2146 ± 193 | 10 ± 1 | 165 ± 11 |
| | 60 | 3.61 | 582 ± 8 | 1465 ± 172 | 36 ± 1 | 616 ± 63 | 59 ± 0 | 1333 ± 144 | 105 ± 3 | 5526 ± 564 | 12 ± 0 | 335 ± 38 |
| | 90 | 3.77 | 666 ± 2 | 1499 ± 163 | 56 ± 2 | 660 ± 73 | 57 ± 3 | 1431 ± 161 | 186 ± 13 | 8063 ± 604 | 12 ± 1 | 366 ± 33 |
| 180 | 30 | 4.05 | 487 ± 18 | 923 ± 50 | 98 ± 2 | 580 ± 14 | 59 ± 2 | 1264 ± 33 | 670 ± 85 | 8852 ± 139 | 21 ± 2 | 304 ± 11 |
| | 60 | 4.22 | 439 ± 39 | 471 ± 13 | 158 ± 2 | 416 ± 12 | 105 ± 0 | 1621 ± 58 | 2181 ± 91 | 9678 ± 222 | 62 ± 4 | 334 ± 31 |
| | 90 | 4.38 | 205 ± 6 | 247 ± 29 | 156 ± 6 | 426 ± 50 | 168 ± 8 | 1655 ± 172 | 2771 ± 58 | 7868 ± 696 | 93 ± 22 | 362 ± 43 |
| 200 | 30 | 4.60 | 66 ± 4 | 73 ± 7 | 145 ± 1 | 149 ± 26 | 336 ± 13 | 1098 ± 115 | 1700 ± 59 | 1630 ± 49 | 105 ± 3 | 146 ± 19 |
| | 60 | 4.81 | 5 ± 2 | 0 ± 0 | 34 ± 4 | 47 ± 4 | 233 ± 12 | 640 ± 28 | 116 ± 16 | 172 ± 13 | 37 ± 3 | 68 ± 9 |
| | 90 | 4.97 | 1 ± 0 | 0 ± 0 | 7 ± 2 | 15 ± 5 | 135 ± 7 | 393 ± 26 | 40 ± 6 | 77 ± 3 | 18 ± 5 | 29 ± 1 |

M: Monomeric. T: Total.

Figure 1 shows the summed concentrations of C5 and C6 sugars for the different LHW pretreatment conditions. Most of the performed extractions have error percentages below 13%, except for monomeric mannose at 180 °C for 90 min (23%) and 200 °C for 90 min (26%), monomeric arabinose at 200 °C for 60 min (33%) and 200 °C for 90 min (26%), and monomeric/total galactose (both 33%) at 200 °C for 90 min. However, the average value at these conditions is low and close to the detection limit.

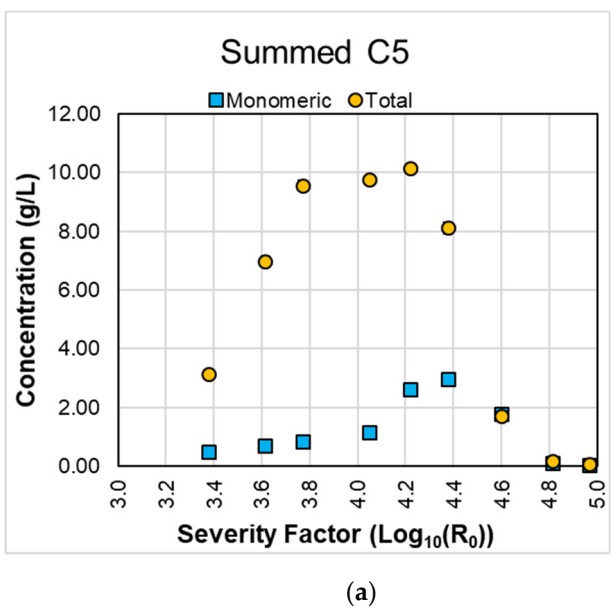

(**a**)

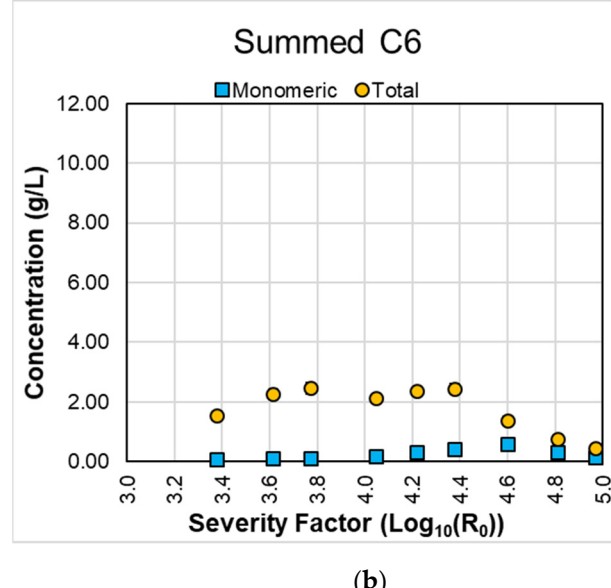

(**b**)

**Figure 1.** *Cont.*

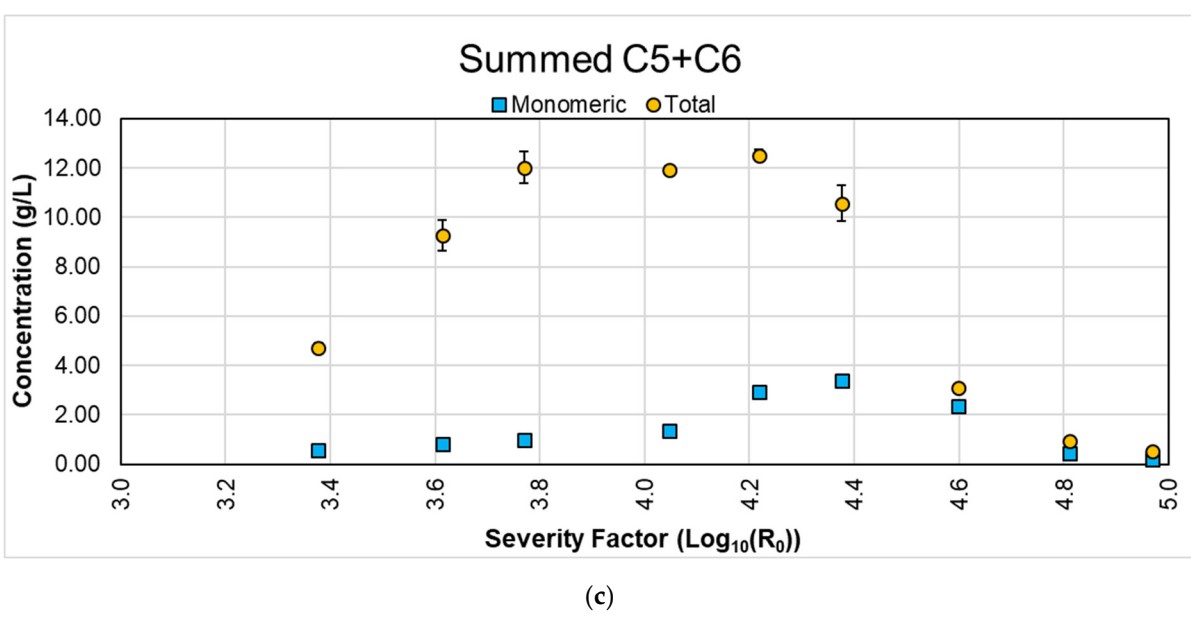

(**c**)

**Figure 1.** Summed concentrations of C5 and C6 sugars for the different LHW pretreatment conditions. Summed C5 (**a**), C6 (**b**), and C5 + C6 sugars (**c**).

LHW is a process mainly focused on the hydrolysis of hemicellulose. Thus, the expected concentrations of C5 sugars should be higher than that of C6 sugars. This was confirmed by the results presented in Figure 1a,b. Besides, in all conditions (except the extractions at 200 °C), the hydrolyzed sugars mainly consisted of oligomers (as observed by the difference between the blue bar corresponding to the monomeric sugars and the green bar for total sugars in Figure 1). Three conditions reached very similar concentrations of total sugars (~12.2 g/L): 160 °C for 90 min, 180 °C for 30 min, and 180 °C for 60 min. This same trend was observed for the total C5 sugar concentration (~10.0 g/L). After the last condition (180 °C for 60 min), the total sugar and C5 sugar concentrations decreased, converting the sugars into degradation products. This will be further discussed in the following section covering the degradation products. The total C6 sugar concentration reached similar concentrations in the range of severity factors between 3.61 (160 °C for 60 min) and 4.38 (180 °C for 90 min).

We found similar trends when comparing these results with other works published in the literature. Huang et al. (2017) evaluated isothermal LHW conditions for wheat straw, between 140–220 °C for 40 min. Within this temperature range, the values at 160, 180, and 200 °C for 40 min can be compared with those obtained in this published work: severity factors are similar (3.37, 3.96, and 4.55, respectively). They reported monomeric xylose concentrations of 1.3, 0.6, and 0.2 g/100 g of wheat straw and oligomeric xylose concentrations of 0.3, 4.9, and 2.0 g/100 g of wheat straw, respectively [27]. These values are within the same ranges found in our work. Carvalheiro et al. (2009) also evaluated isothermal LHW conditions, and comparable severity factors were achieved at 200 °C and 26 min: monomeric xylose and glucose concentrations of, respectively, 1.6 and 0.8 g/L were reported, which are within the ranges obtained in our work [19].

In terms of the usability of the obtained sugars, the obtained concentrations of monomeric sugars in the present work were considerably low, meaning that the hydrolysate should not be directed to microorganisms only able to metabolize monomeric xylose and glucose. This requires further steps of hydrolysis of the oligomeric sugars into monomers. Instead, the hydrolysate could be used for fermentation with microorganisms metabolizing monomeric and oligomeric sugars. Some studies have been carried out to produce, e.g., polyhydroxybutyrate and tetraether lipids [18,28]. In this context, three of the studied conditions reached similar levels of total sugars to be potentially used, namely, 160 °C for 90 min, 180 °C for 30 min, and 180 °C for 60 min. However, due to the different severity

factors, each set of conditions has a different profile of degradation products, restricting their usage, as further explained in the following section.

### 3.3. Degradation Products

Figure 2 shows the concentrations of degradation products reached under each studied condition. The analysis of these components is relevant as they may condition future usages of the hydrolysates as a substrate for fermentation. We observed a steady increase in the concentration of acetic acid until a severity factor of 4.60 (200 °C for 30 min). Acetic acid is produced from the release of the acetyl groups connected to the hemicellulose matrix. This suggests that hemicellulose is hydrolyzed until the above severity factor is reached. However, even with the maximum acetic acid concentration reached in this work, ~3.2 g/L, the concentrations are below reported thresholds at which acetic acid inhibits microorganism growth (e.g., 7.5–15.0 g/L for yeast growth [29,30]).

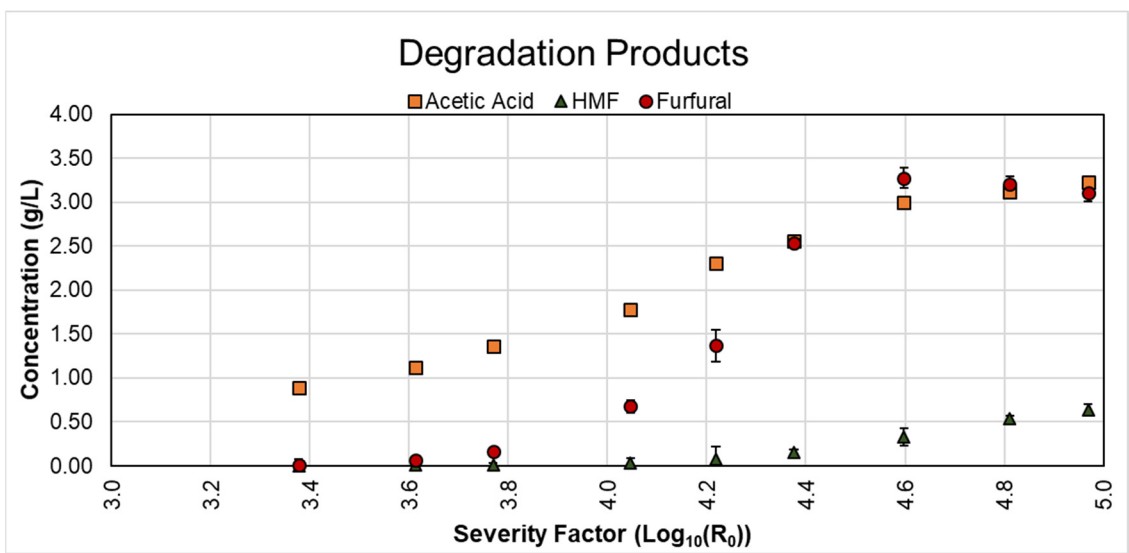

**Figure 2.** Concentration of degradation products for the different LHW pretreatment conditions.

Regarding HMF, significant concentrations only started to be reached at a severity factor of 4.22 (180 °C for 60 min), and it continued increasing steadily until reaching 0.5–0.6 g/L for the two final conditions studied in this work. This severity factor coincided with the decrease of C6 sugars, as observed in Figure 1b. According to HMF results from the dehydration of C6 sugars, meaning those above a severity factor of 4.22, the produced C6 sugars were degraded into HMF. Previous studies reported that the presence of HMF increases the duration of the lag phase for yeast [31], and significant growth inhibition (45%) has been observed at concentrations of 0.5 g/L [32,33]. This value was achieved in the present study using 200 °C for 60 min (severity factor: 4.81) or higher.

Regarding furfural, significant concentrations started to be reached at a severity factor of 3.77 (160 °C for 90 min), and it continued increasing steadily until reaching ~3.2 g/L at a severity factor of 4.60 (200 °C for 30 min). Its concentration remained nearly constant for the remaining conditions. Regarding furfural from C5 sugars dehydration, Figure 1a shows that, after a severity factor of 4.22 (180 °C for 60 min), the C5 concentration decreased, meaning that no C5 sugars were being hydrolyzed but were being degraded into furfural. In terms of inhibition thresholds, different authors reported growth inhibition and biomass yield decreases for yeast and bacteria with furfural concentrations between 0.5 and 1.5 g/L and higher [31,33,34]. This is a critical indicator to choose the conditions with furfural levels behind this concentration. Accordingly, the maximum severity factor that can be reached is 4.22 (180 °C for 60 min) in a conservative scenario taking the upper limit of the furfural inhibition threshold, or below 4.05 (180 °C for 30 min) for a stricter scenario

taking the lower limit. These results are comparable with the values reported for wheat straw by Huang et al. (2017): furfural concentrations of 0.4, 0.1, and 0 g/L and acetic acid concentration of 2.1, 1.9, and 1 g/L at 160, 180, and 200 °C for 40 min were observed. At 180 °C for 60 min, concentrations of 0.2 and 1 g/L, for furfural and acetic acid, respectively, were attained [27].

### 3.4. Lignin

Generally, publications analyzing LHW present the results regarding sugar production and degradation products as a core section and cover other topics, mainly the resulting solid's enzymatic digestibility. However, creating an integral perspective that provides information on the lignin hydrolyzed by LHW is still missing, and this is essential to further design pretreatment stages focused on the lignin fraction. This section will now cover the obtained lignin hydrolysis yields under the different studied conditions, as shown in Figure 3.

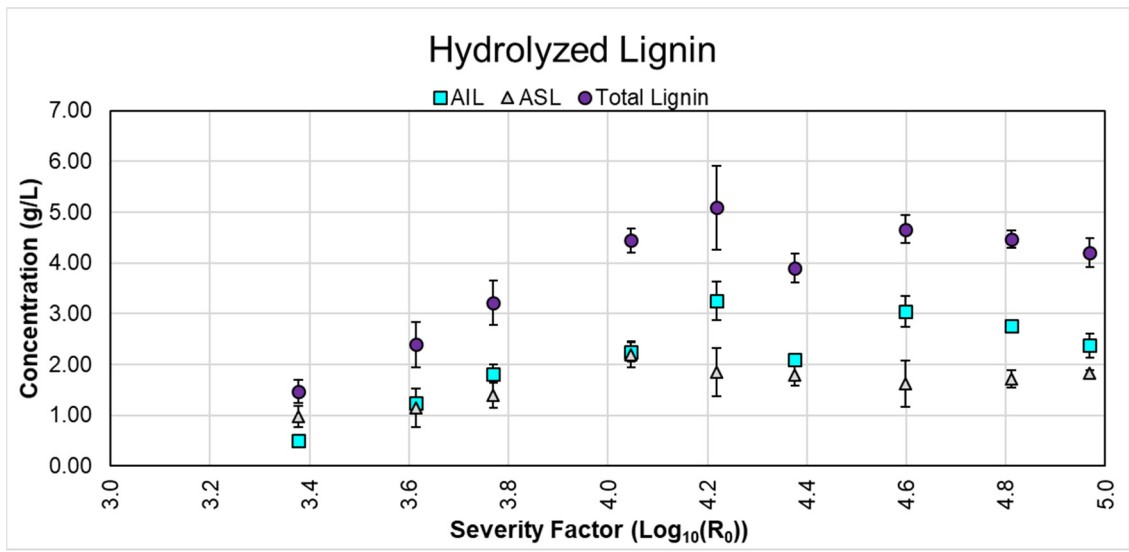

**Figure 3.** Concentration of hydrolyzed lignin for the different LHW pretreatment conditions tested. AIL: Acid Insoluble Lignin; ASL: Acid Soluble Lignin. Total lignin corresponds to the sum of AIL and ASL.

The goal regarding lignin should be to maintain it as much as possible in the solid fraction. We can observe that the total lignin increases steadily until reaching a maximum value between severity factors of 4.05 and 4.22 (180 °C for 30 min and 180 °C for 60 min, respectively) and then decreases. AIL followed this same trend, reaching a maximum value of 3.2 g/L at the same severity factor at which total lignin reached the maximum. A maximum lignin concentration with an increasing severity factor for LHW is explained because after a specific set of temperature and time (namely, a value of severity factor), the solubilized lignin re-condensates and would not be soluble anymore. This means the lignin would remain as part of the solid fraction. Therefore, once the LHW extraction is finished, the resulting mixture is pressed, and the extract is further centrifuged (as described in the methodology), the lignin would go to the solid fraction and the respective precipitate from centrifugation. Therefore, when the extract is characterized, even though the solid was submitted to a higher severity factor, this does not implicate that more lignin would remain solubilized in the extract. Compared to the initial value of lignin in the raw material, LHW conditions at 160 °C reached lignin solubilization yields between 10% and 15%, at 180 °C between 26% and 35%, and 200 °C between 16% and 29%. Based on this, the extractions carried out at 160 °C fulfilled the purpose of solubilizing less lignin. It is important to stress again that LHW is a process not meant to solubilize and valorize lignin but hemicellulose.

Therefore, the next step should be to couple the results from the three analyzed conditions, which will be done in the following section.

### 3.5. Integrated Analysis of Sugars, Degradation Products, and Lignin

This section focuses on integrating the results obtained in the previous sections. For this purpose, we will use the most relevant values from each section. To evaluate the extraction of sugars, we will take as base the total summed C5 + C6 sugars, as it was shown that the potential use of the LHW hydrolysate as fermentation substrate should be for microorganisms metabolizing both monomeric/oligomeric C5 and C6 sugars. We performed the analysis for the three conditions that reached higher total summed C5 + C6 sugars (160 °C for 90 min, 180 °C for 30 min, and 180 °C for 60 min). We observed that furfural was the component that reached critical concentrations at a lower severity factor when analyzing the degradation products. Therefore, we will use the concentration of furfural as the decision criteria. Regarding lignin extraction, we will use the total lignin extracted, given that this value reflects the entire amount removed from the solid matrix at given conditions. Figure 4 shows the combined results of total sugars, furfural, and total lignin for the three sets of conditions that allowed the higher total concentration of sugars.

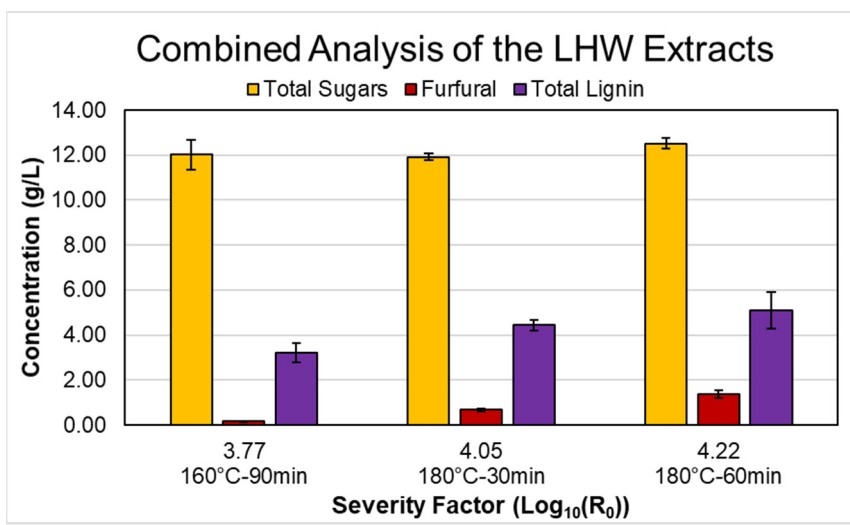

**Figure 4.** Combined analysis of sugars, degradation products, and lignin for the LHW pretreatment conditions. Total sugars correspond to the total summed C5 and C6 sugars. Total lignin corresponds to the sum of AIL and ASL.

Based on the primary goal of this study, to reach a high sugar concentration and low concentration of degradations products and extracted lignin, it is clear that performing the pretreatment at 160 °C for 90 min (severity factor of 3.77) rendered the best overall results. Sugar production reached 98% of the maximum achieved value (at 180 °C for 60 min), and on the other hand, furfural and lignin corresponded only to 11.8% and 43.9%, respectively. Therefore, the produced hydrolysate is below the inhibition threshold, and most of the lignin remains in the solid phase for further valorization. Based on the amount determined in the liquid extract for the extraction at 160 °C for 90 min and the wheat straw composition, we calculated the solid composition, assuming that all of the glucose corresponded to cellulose and the other characterized sugars corresponded to hemicellulose as shown in Figure 5. As observed, the remaining solid still contains around 80% of the lignin and 61% of the hemicellulosic sugars contained in the initial feedstock. Cellulose increased from 45.2 to 53.4 wt %, while hemicellulose and lignin decreased from 32.6 to 24.6 wt % and 22.2 to 22.0 wt %, respectively. This LHW solid could be used, for example, to replace lignin-rich pulps used in papermaking (e.g., mechanical pulps). On the other hand, wheat straw has been extensively studied for lignin production, and previous studies have shown

that performing LHW before a delignification stage improves the extraction yield [5,15]. Therefore, LHW solids from wheat straw can be used further for lignin extraction, and this study case will be developed further in the next section.

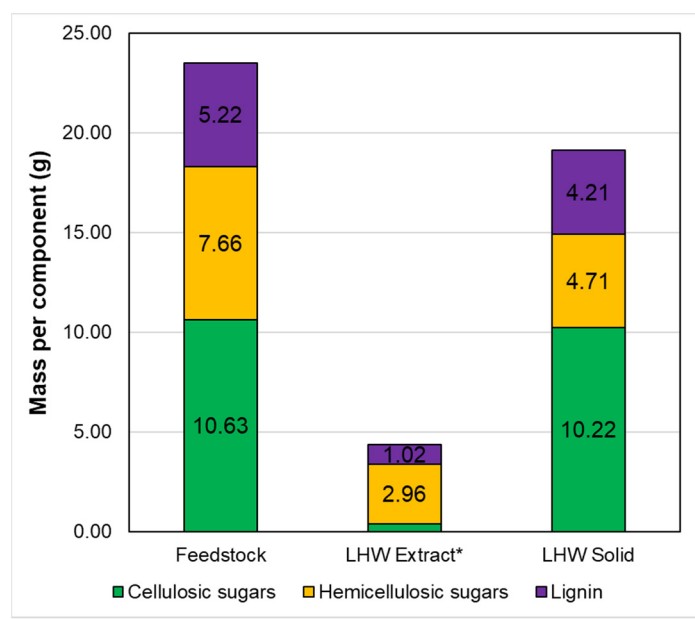

**Figure 5.** Absolute mass balance for cellulose, hemicellulose, and lignin in the feedstock, LHW extract, and the solid leaving the LHW stage for the LHW carried out at 160 °C for 90 min. * Cellulosic sugars for the LHW Extract are 0.41 g.

### 3.6. Future Outlook: Study Case—Theoretical Valorization of the Solids for Lignin Hydrolysis

Considering the previous results and following the strategy proposed by Serna-Loaiza et al. (2021), we decided to evaluate theoretically the sequential hydrolysis of hemicellulose and lignin using LHW followed by Organosolv (OS) [15]. These authors reached hydrolyzed lignin concentrations of ~7 g/L in the standalone Organosolv, and this value increased to ~11 g/L when LHW was performed before the OS. We evaluated the scenario of subsequent valorization of the solids remaining after the LHW using the latter concentration (11 g/L). The conditions chosen for the LHW stage were 160 °C for 90 min. Figure 6 shows the mass balances for the LHW followed by OS (LHW→OS).

Based on the composition and moisture of the solid leaving the LHW stage, we calculated the solvent required for the OS stage (Figure 6a). We used the same proportions proposed by Serna-Loaiza et al. (2021) (S/L ratio of 1 g dry matter per 1 mL of solvent, 60 wt % ethanol as solvent), and the reached concentrations of sugars, degradation products, and lignin [15]. We calculated the composition of the lignin extract and the respective solid with this information. We proceeded to calculate the mass balance using the density of the extracts (1.01 g/mL and 0.9 g/mL for sugar and lignin extracts, respectively). We provide all the information related to each extraction's mass balance in the Supplementary Materials. These results correspond to a valorization outlook of the solids after LHW pretreatment, as the specific values obtained from the OS extraction of the LHW solid depend on variables and conditions that can result only from carrying out the specific experiments.

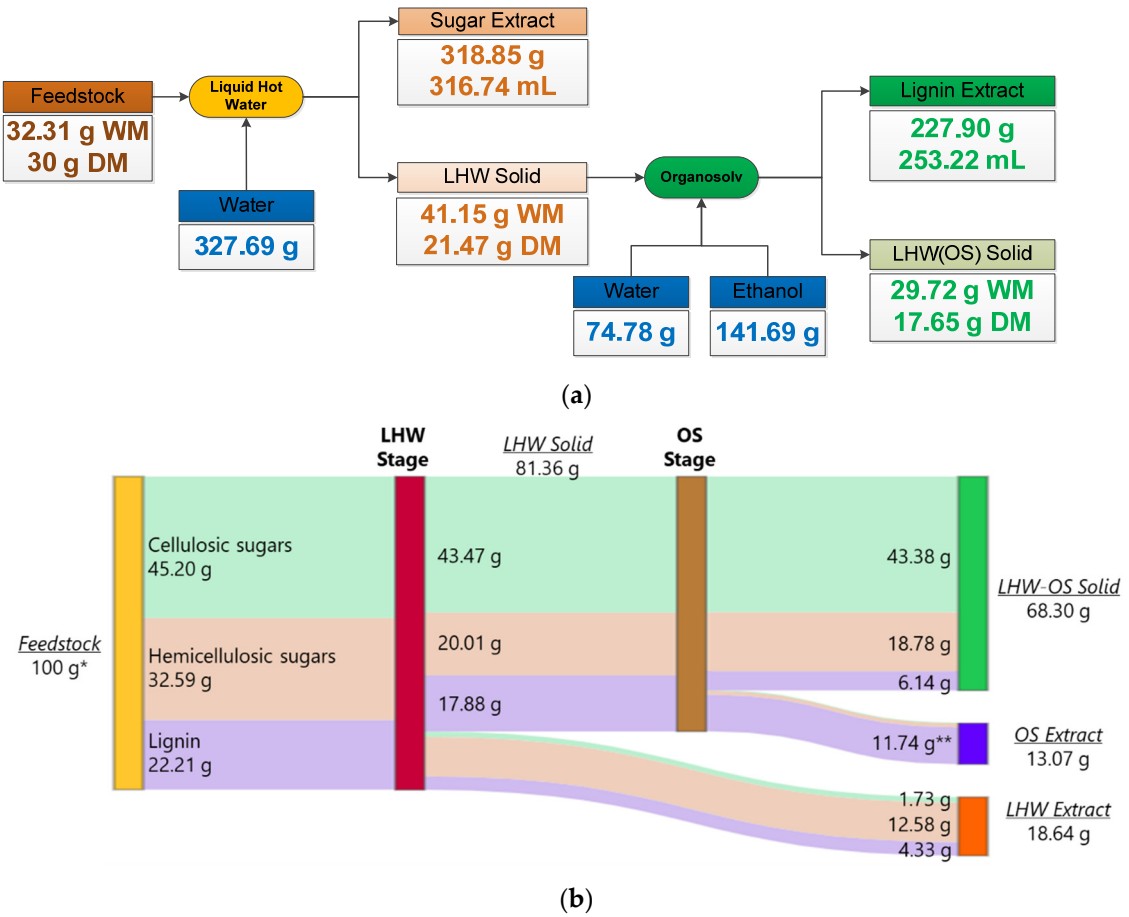

**(a)**

**(b)**

**Figure 6.** Mass balances and distributions of the streams for the theoretical LHW-OS combination. (**a**) General mass balance of the LHW and OS stages. (**b**) Distribution of cellulose, hemicellulose, and lignin along the stages of the theoretical LHW-OS combination. * Feedstock composition only corresponds to the amount of cellulose, hemicellulose, and lignin. ** The amounts of cellulose and hemicellulose in the OS extract are below 1.2 g and cannot be visualized adequately in the figures.

Figure 6b shows the distribution of cellulose, hemicellulose, and lignin along the LHW and OS stages. We can observe an excellent distribution of the lignocellulosic components along the different streams; an LHW extract, an OS extract, and a final solid mainly composed of hemicellulosic sugars, and cellulose, respectively. Cellulose conversion into sugars corresponded only to 5%, meaning that the final solid preserves around 95% of the initial cellulose. Regarding the LHW extract, 38.6% of hemicellulosic sugars went to this stream, while 57% of these sugars remained in the LHW-OS solid. Finally, regarding lignin, 52.9% of this component is extracted in the OS extract and 33.6% remain in the LHW-OS solid. This scenario provides an outlook on the solid fraction's possible use and further valorization after the LHW stage. Cellulose represents ~50% of the solid mass, and as previously analyzed, 72% of the lignin has been removed. This shows a potential application in materials (pulp and paper).

This combinatorial pretreatment (LHW followed by OS) allows obtaining three intermediate products streams (a sugar-rich and a lignin-rich extract and a cellulose-enriched solid), increasing the possible economic outputs of the process. On the other hand, these two technologies as standalone processes entail significant energy consumption associated with the heating and recovery of the solvent of the OS stage, which implicates a challenge to the feasibility of the process. However, the actual combinatorial pretreatment may implicate an improvement for the OS-standalone process. By performing LHW first, around 28% of the initial mass is solubilized (as observed in Figure 6a,b), which implicates lower solvent requirements and fewer liquid extracts to be processed in the downstream, which could

compensate for the inclusion of a further step. However, this hypothesis requires further investigation to evaluate this strategy's technical and environmental benefits.

## 4. Conclusions

This study tested different temperatures and times for the LHW pretreatment of wheat straw and determined the extent of lignin solubilization, in addition to the determination of sugars and degradation products. We showed that the LHW at 160 °C for 90 min (severity factor of 3.77) allowed the best extraction of components, reaching a total concentration of sugars of ca. 12 g/L, and 0.2, 0.01, and 1.4 g/L for furfural, HMF, and acetic acid, respectively. A lignin concentration of 2.2 g/L was also attained. By including the analysis of lignin hydrolysis into the standard sugar and degradation products in LHW, it was possible to make a more integrated decision to valorize the different lignocellulosic components of wheat straw. The theoretical study case presented for lignin extraction in a subsequent OS stage indicates the potential of lignin extraction and valorization of the final solid for real, sound applications.

**Supplementary Materials:** The following are available online at https://www.mdpi.com/article/10.3390/su14010362/s1, Supplementary Material S1—Summary of the characterization of the LHW extracts, S2—Mass balance of the LHW stage, S3—Mass balance for the LHW-OS combination, S4—Streams balance for the LHW-OS combination.

**Author Contributions:** Conceptualization, S.S.-L., L.D.-S. and A.F.; investigation, S.S.-L. and M.D.; writing—original draft preparation, S.S.-L.; writing—review and editing, M.D., L.D.-S., C.C.C.R.d.C. and A.F.; supervision, C.C.C.R.d.C. and A.F. All authors have read and agreed to the published version of the manuscript.

**Funding:** This research received no external funding.

**Acknowledgments:** The authors acknowledge TU Wien for funding the Doctoral College "Bioactive", under which this research was performed, and TU Wien Bibliothek for the financial support through its Open Access Funding Programme. The authors are grateful to the ERASMUS + Program of Higher Education for Traineeships of the European Commission for financing the research stay of Manuel Francisco Nunes Dias in TU Wien.

**Conflicts of Interest:** The authors declare no conflict of interest.

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
