# Peer review of "Integral Analysis of Liquid-Hot-Water Pretreatment of Wheat Straw: Evaluation of the Production of Sugars, Degradation Products, and Lignin"

_sustainability, doi:10.3390/su14010362_

Round 1

Reviewer 1 Report

Reviewer Comments:

This manuscript applied a liquid-hot-water method to pretreat lignocellulosic biomass. I think this is a meaningful work, but there are also some areas need to be improved. This manuscript could be acceptable only after major modifications.

Comments:

  1. In Lines 323-324, authors described that the solid mass was composed of ~50% of cellulose, and 72% of the lignin has been removed. While the Figure 5 (a) shows that 5.22 g lignin was contained in feedstock, and still remained 4.21 g of lignin in the LHW solid. Thus, did authors ensure that 72% of lignin was removed?
  2. In Lines 51-61, many precious studies related with combination of pretreatments were listed, please add the advantages and disadvantages of these methods. Did the authors use methods that have been used in other studies? Has there been an improvement in methodology?
  3. Please add more comparison in the manuscript, which may highlight the advantages of this method.
  4. Please add the potential applications of the lignocellulosic biomass after using the LHW pretreatment method.
  5. Please add some discussion about the economic challenges and opportunities.

Reviewer 2 Report

Valorisation of lignocellulosic biomass has high importance int he field of renewable energy generation and waste handling and utilization, respectively. But, because of the complex physicochemical structure of LCB, the utilization technologies need efficient pre-treatment. LHW is a promising method for lignin, cellulose and hemicellulose hydrolysis, as well. Detailed investigation of the effect of process parameters and determination and analysis of degradation products can provide interesting information for the readers. Therefore the topic of the manuscript is relavant and provides useful information for the readers. The manuscript is generally well structured. Introduction section summarizes well the general characteristics and applicability of LHW, and the research motivations. Applied characterization methods are adequate to the sample characteristics. Description of process condition and characterization method is clear. Manuscript contains interesting and results that are valuable not just for the science but also for the practice. Experimental results are represented well in figures and tables and discussed in details with relevant references.

Comments, suggestions:

I suggest the authors to highlight why use wheat straw as a ’test’ raw material for LHW.

In my opinion sentence ‘In this section, we will analyze the respective collected data’ in line 133 is unnecessary.

Please give reference for establishment in line 154-158.

Please improve the visibility of axis title and text in Figure 1-3, and Fig. 5.a.

Please give DOIs for all references (if available).

Round 2

Reviewer 1 Report

I think the revised manuscript meets the requirments of the journal publication.